# Therapeutic Education for Safer Rheumatologic Care: A Scoping Review to Map Evidence on Infection Prevention

**DOI:** 10.3390/nursrep15120431

**Published:** 2025-12-04

**Authors:** Khadija El Aoufy, Camilla Elena Magi, Maria Ramona Melis, Cristiana Caffarri, Giovanni Civile, Elena Daffini, Eleonora Loss, Helena Ortis, Antonella Rinaldi, Claudia Zonca, Stefano Bambi, Laura Rasero

**Affiliations:** 1Department of Health Sciences, University of Florence, 50134 Florence, Italy; khadija.elaoufy@unifi.it (K.E.A.); stefano.bambi@unifi.it (S.B.); l.rasero@unifi.it (L.R.); 2Careggi University Hospital, 50134 Florence, Italy; mariaramona.melis@unifi.it; 3Department of Medicine and Surgery, University of Parma, 43121 Parma, Italy; cristiana.caffarri1@studenti.unipr.it (C.C.); giovanni.civile@studenti.unipr.it (G.C.); elena.daffini@studenti.unipr.it (E.D.); eleonora.loss@studenti.unipr.it (E.L.); helena.ortis@studenti.unipr.it (H.O.); antonella.rinaldi1@studenti.unipr.it (A.R.); claudia.zonca@studenti.unipr.it (C.Z.)

**Keywords:** rheumatic and musculoskeletal diseases (RMDs), disease-modifying anti-rheumatic drugs (DMARDs), infection prevention, therapeutic education, patient education, nursing

## Abstract

**Background**: Patients with Rheumatic and Musculoskeletal Diseases (RMDs) who are treated with Disease-Modifying Anti-Rheumatic Drugs (DMARDs) face an increased risk of infections. Therapeutic education is often considered a valuable strategy to support preventive behaviors, but its actual impact remains uncertain. **Objectives**: This scoping review aims to examine how therapeutic education contributes to infection prevention in patients with RMDs receiving DMARDs, with attention to its potential benefits, limitations, and relevance in clinical practice. **Methods**: Following the PRISMA-ScR framework, we searched PubMed, CINAHL, EMBASE, and Web of Science for primary studies published between January 1990 and December 2024 in English or Italian language. Eligible studies involved adult patients with rheumatic diseases treated with DMARDs who had received some form of therapeutic education. **Results**: Among 1591 records, only 4 studies met the inclusion criteria. These studies emphasized the value of promoting preventive behaviors to minimize treatment-related infections. Therapeutic education was associated with increased patient awareness and adherence, especially when supported by multidisciplinary healthcare teams. However, several barriers—such as limited health literacy and socioeconomic challenges—affected access and effectiveness. **Discussion and Conclusions**: While existing studies support the potential of therapeutic education and patient education in general, the small number of relevant studies and the variation in approaches limit strong conclusions on the impact of patient education on reducing or preventing risk infection in the field of rheumatology in DMARD-treated patients. Moreover, several papers pointed out how digital tools and telemedicine are promising ways to expand access and improve adherence, particularly for underserved populations. Thus, further research should explore standardized, inclusive and interdisciplinary strategies—potentially incorporating digital tools—to improve prevention and ensure equitable access to educational interventions.

## 1. Introduction

Rheumatic and Musculoskeletal Diseases (RMDs) encompass a heterogeneous group of nearly 200 conditions that primarily affect the joints and connective tissues, but may also involve virtually any organ system [1]. RMDs are typically chronic in nature, marked by alternating periods of remission and flare-ups, and have a profound impact on both physical and psychosocial well-being [2]. RMDs are influenced by a strong interplay between genetic predisposition and environmental factors [3]. Elements such as infections, smoking, diet, stress, and lifestyle can interact with genetic susceptibility, shaping disease onset, severity, and clinical expression [4].

Effective management of these diseases relies on timely diagnosis, early initiation of treatment, and the prevention of complications [5]. The pharmacological management of RMDs has undergone major advances in recent decades, evolving from a predominantly symptomatic approach, based on analgesics and Nonsteroidal Anti-Inflammatory Drugs (NSAIDs), to strategies aimed at directly modifying disease activity through the use of Disease-Modifying Anti-Rheumatic Drugs (DMARDs) [6].

First-line management typically involves the use of NSAIDs and corticosteroids, which provide rapid relief of pain and inflammation. However, prolonged use can cause side effects such as the increase in the risk of stomach ulcer/gastrointestinal disturbances and the risk of infections, respectively. Second-line treatment is centered on DMARDs, which aim to slow or halt disease progression but generally require more time to achieve clinical efficacy [7]. Within this category, several classes can be distinguished: (i) Conventional synthetic DMARDs (csDMARDs) exert broad immunosuppressive effects to reduce inflammation; (ii) targeted synthetic DMARDs (tsDMARDs), such as Janus kinase (JAK) inhibitors, selectively block intracellular signaling pathways involved in immune activation; (iii) Biologic DMARDs (bDMARDs) were produced using recombinant biotechnology, acting on specific molecular targets (e.g., cytokines or immune cells); (iv) Biosimilar DMARDs (bsDMARDs) are highly similar versions of reference biologics, approved after rigorous demonstration of comparability in terms of efficacy, safety, and immunogenicity [6]. This therapeutic evolution has significantly improved clinical outcomes and quality of life, but has also introduced new safety concerns, particularly related to infection risk [5,6].

However, these therapeutic benefits are accompanied by potential risks and adverse effects. While csDMARDs exert broad immunosuppressive activity, bDMARDs are designed to selectively target key mediators of the inflammatory cascade, including tumor necrosis factor (TNF), Janus kinase (JAK), B cells, and various interleukins (e.g., IL-1, IL-6) [6]. When prescribing these agents, it is essential to account for patient-specific factors such as age, concomitant immunosuppressive therapies, and drug dosage, all of which can influence susceptibility to infection [7]. High biologic dosages, for example, have been associated with a nearly 90% increase in infection risk, particularly among patients treated with TNF inhibitors or those concurrently receiving corticosteroids and csDMARDs [8,9].

Given these concerns, both European Alliance of Associations for Rheumatology (EULAR) and the American College of Rheumatology (ACR) recommend conducting pre-biologic screening and ensuring that patients receive appropriate vaccinations and information before initiating bDMARDs’ therapy [10,11].

Nevertheless, patients treated with DMARDs often lack adequate awareness of infection-related risks and preventive measures, underscoring the importance of effective patient education and shared decision-making [12]. Within a broader strategy for managing RMDs, patient education (PE) plays a crucial role. PE can be delivered in different formats: individually, in groups, or through blended approaches, and increasingly employs online or digital tools such as videos, multimedia resources, and mobile applications. Regardless of the format, the process must remain patient-centered, rooted in the principle of “shared decision-making” [13]. Among the educational models developed for RMDs, interactive one-on-one sessions and combined educational programs lead by specialist nurses currently provide the strongest level-one evidence for improving patient outcomes and disease management [13]. Considering the increasing use of bDMARDS in adult patients with RMDs, continuous monitoring of their safety profile is crucial [14]. Equally important is ensuring that patients are adequately informed about infection-related risks and preventive strategies [5,11]. Thus, this scoping review aims to map and analyze the existing literature on therapeutic education strategies targeting infection risk in patients with RMDs receiving anti-rheumatic drugs.

## 2. Materials and Methods

### 2.1. Study Design

This scoping review was conducted according to the five-stage methodological framework proposed by Arksey and O’Malley [15], which includes the following: (1) identifying the research question, (2) identifying relevant studies, (3) selecting the studies, (4) charting the data, and (5) collating, summarizing, and reporting the results. Additional methodological enhancements recommended by Levac et al. [16] were incorporated, including iterative refinement of the search strategy and comprehensive data charting procedures. This review followed the PRISMA-ScR (Preferred Reporting Items for Systematic Reviews and Meta-Analyses extension for Scoping Reviews) [17].

### 2.2. Research Question

The research question was formulated using the PCC (Population: adults aged 18 years or older diagnosed with RMDs and treated with anti-rheumatic drugs; Concept: therapeutic education and their reported outcomes; Context: infection prevention and safety within anti-rheumatic therapy) framework. The research question was: What therapeutic education interventions have been studied to address infection prevention in adults with RMDs treated with anti-rheumatic drugs, and what outcomes have been reported?

### 2.3. Information Sources and Search Strategy

A comprehensive literature search was conducted across four electronic databases: PubMed, CINAHL, EMBASE, and Web of Science (combining controlled vocabulary and free-text terms). The search was restricted to studies published between January 1990 and October 2024. Gray literature was not systematically searched due to feasibility and focus on peer-reviewed evidence for methodological rigor. A reverse reference search of the included studies was performed to identify any additional eligible articles, but it did not yield further records meeting the inclusion criteria. An example of the PubMed search strategy is reported in Appendix A.

### 2.4. Study Selection

Eligibility criteria were defined a priori. Studies were included if they were primary research articles (quantitative, qualitative, or mixed methods designs) involving adult populations (≥18 years) with rheumatic diseases treated with anti-rheumatic drugs. Studies were limited to English and Italian publications, reflecting the language proficiency of the review team. The time frame for publication was from January 1990 to December 2024, chosen to capture the evolution of anti-rheumatic therapies in rheumatology. Exclusion criteria were studies involving pediatric populations (<18 years) or mixed populations where adult data could not be extracted separately, secondary literature such as systematic reviews or meta-analyses, narrative reviews, letters to the editor, editorials, conference abstracts, dissertations, and theses. The selection process followed two stages: initial title and abstract screening and subsequent full-text assessment against the inclusion criteria. Two reviewers independently conducted both phases, resolving discrepancies by discussion or consulting a third reviewer when necessary.

### 2.5. Data Extraction

Data extraction was carried out using a standardized MS Excel^®^ form specifically designed for this review. Extracted information included study characteristics (i.e., author, year, country, study design, setting), population details (i.e., sample size, age, gender, type and duration of rheumatic disease, treatment regimen, comorbidities), intervention features (i.e., type of therapeutic education, theoretical framework, duration, frequency, delivery method, healthcare providers involved, and any measures of intervention fidelity), outcome measures (i.e., knowledge scores, self-efficacy, adherence rates, and quality of life), and key findings related to therapeutic education in relation to infection risk among individuals with RMDs treated with anti-rheumatic drugs.

### 2.6. Quality Assessment

Although quality appraisal is not mandatory in scoping reviews, it was conducted to provide contextual insight into the robustness of the evidence base. Thus, the methodological quality of the included studies was appraised using the Critical Appraisal Skills Program (CASP) tools, with specific checklists applied according to the study design. The appropriate CASP checklist (for randomized controlled trials, cohort, or cross-sectional studies) was used for each study. Each study was scored based on the number of “yes” responses, and overall quality was categorized as high (≥7), moderate (4–6), or low (≤3). Two reviewers independently assessed study quality, resolving any disagreements by consensus.

### 2.7. Data Synthesis

Extracted data were synthesized descriptively to map the extent, range, and nature of the available evidence. Study characteristics and intervention features were summarized in tables and narrative form. Outcomes related to infection risk, knowledge, self-efficacy, and adherence were grouped thematically to identify patterns and research gaps. Quantitative findings were summarized using descriptive statistics, while qualitative data were narratively integrated.

## 3. Results

A total of 1591 records were identified across four electronic databases. After removing 8 duplicates, 1583 records were screened, of which 4 studies met all inclusion criteria and were included in this scoping review (Figure 1).

### 3.1. Quality of Included Studies

The methodological quality of the included studies was appraised using the CASP checklists, applying the specific tools for RCTs and for cohort or cross-sectional studies. A detailed evaluation of each study according to the CASP domains is presented in Table 1. Overall, the methodological quality was judged as good-to-moderate. Among the two observational studies (cohort and cross-sectional), both met most of the quality criteria, demonstrating robust methodological standards and adequate reporting. The RCT also showed good methodological quality, fulfilling ten out of twelve CASP items. Conversely, the quasi-experimental study displayed moderate quality, primarily due to the absence of randomization and control procedures, and limited clarity regarding the description of the intervention.

### 3.2. Characteristics of Included Studies

The four studies included in this scoping review were published between 2020 and 2022, all conducted in Europe. One study was carried out in France, one in Germany, one in the United Kingdom, and one was a multinational study involving Spain, the United Kingdom, Germany, France, and Sweden. All studies focused on patients with inflammatory RMDs treated with bDMARDs or tsDMARDs. Study designs included one randomized controlled trial, two cross-sectional studies, and one cohort study, with sample sizes ranging from 127 to 975 participants. The majority of participants were female and middle-aged to older adults. Two studies explicitly involved multidisciplinary teams comprising rheumatologists and nurses, emphasizing the role of collaborative patient education in infection prevention and management. The interventions or initiatives explored varied across studies: a nurse-led educational program to improve safety competencies among patients with inflammatory arthritis [20]; a patient alert card system to enhance infection awareness and reporting in individuals treated with abatacept [18]; an infection screening and vaccination audit among patients receiving biologic therapy [19]; and a risk stratification and targeted information campaign to mitigate COVID-19 infection risk [21]. Detailed study characteristics are presented in Table 2.

### 3.3. Educational Interventions and Outcomes in Infection Prevention Among Rheumatology Patients

The included studies explored diverse educational approaches aimed at enhancing infection prevention, vaccination adherence, and safety behaviors among patients with inflammatory rheumatic diseases treated with bDMARDs or tsDMARDs. Two studies focused on informational tools [18,21], one evaluated a structured nurse-led education program [20], and another assessed vaccination-related knowledge and practices [19]. Educational strategies were associated with improvements in patients’ awareness, preventive behaviors, and infection-related outcomes, although the extent of benefit varied across interventions. Informational materials, such as patient alert cards or targeted letters, were linked to higher tuberculosis screening rates (*p* = 0.004) and infection incidences comparable to those of the general population (0.403% vs. 0.397%), suggesting that brief, standardized communication can effectively promote patient vigilance and adherence to preventive recommendations [18,21]. Structured, nurse-led education demonstrated a stronger effect, with intervention participants showing significantly higher safety behavior scores compared to controls (mean biosecure score 81.2 ± 13.1 vs. 75.5 ± 13.0; *p* = 0.016), supporting the value of personalized education and multidisciplinary involvement in improving self-management competencies [20]. Conversely, despite the routine provision of information, persistent gaps were observed in vaccination coverage, with only 55% of patients presenting vaccination cards and low immunization rates associated with higher disease activity and immigrant background [19]. Further details are reported in Table 3.

## 4. Discussion

This scoping review mapped the existing evidence on therapeutic education strategies aimed at infection prevention among patients with RMDs treated with b/tsDMARDs. Four studies were identified, encompassing informational, communicative, and nurse-led interventions. Across the included studies, three categories of educational interventions were identified: informational interventions, tailored communicative strategies, and structured nurse-led therapeutic education programs. Each intervention type was applied in different contexts and targeted infection prevention through specific mechanisms such as improving risk awareness, screening adherence, and self-management. Overall, educational approaches were associated with improved infection awareness and safer health behaviors, although vaccination adherence remained suboptimal, suggesting that knowledge alone may not translate into sustained behavioral change. This is the first synthesis specifically addressing infection-related therapeutic education in rheumatology. The findings emphasize the importance of multidisciplinary, patient-centered approaches and highlight the expanding role of nurses and digital tools in delivering structured and empowering education beyond traditional pharmacologic safety and surveillance frameworks.

Across studies, patient education consistently emerged as a determinant of preventive behaviors and safety awareness. Despite widespread dissemination of information, a persistent gap remained between knowledge and vaccination adherence, with only half of patients meeting national immunization guidelines [19]. This highlights the challenge of translating knowledge into behavior change and the need for structured, interactive education actively engaging patients in infection prevention.

Proactive communication strategies also proved valuable: tailored informational letters to high-risk patients achieved infection rates comparable to the general population, suggesting that personalized communication enhances risk perception and adherence [21]. Likewise, the use of patient alert cards improved tuberculosis screening and reduced infection-related hospitalizations, showing that simple, accessible materials can promote safety awareness and self-management [18].

Structured, nurse-led interventions demonstrated the strongest impact on safety behaviors and coping strategies compared with standard care, confirming the relevance of therapeutic education as part of chronic disease management [20]. Although severe infection rates were unchanged, these findings underline the role of education in strengthening self-efficacy and adherence. These findings are consistent with previous work in chronic disease education showing that interactive, patient-centered educational approaches improve adherence and self-efficacy more effectively than informational strategies alone [22].

Overall, the evidence supports EULAR recommendations to integrate therapeutic education and self-management into routine rheumatology care [23,24]. Multidisciplinary collaboration, particularly involving nurses, facilitates personalized, continuous education addressing both clinical and psychosocial needs. Beyond infection prevention, the broader chronic disease literature confirms that education enhances health literacy, self-efficacy, and adherence, reducing hospitalizations and improving quality of life. The expanding role of nurses and the adoption of digital and tele-educational tools represent promising directions for scalable, patient-centered prevention strategies.

Nonetheless, few studies included in this scoping review have quantified the effect of education on infection outcomes in immunosuppressed patients. Heterogeneity in study design, populations, and outcomes limits comparability across the included evidence. Additionally, no study used standardized outcome frameworks for therapeutic education, and follow-up durations were generally short, restricting the assessment of long-term behavioral change. Finally, the search was limited to studies published in English and Italian due to reviewer language proficiency; this pragmatic constraint may have introduced language bias and led to the omission of relevant studies published in other languages.

Future research should develop standardized frameworks to assess effectiveness, scalability, and cost-efficiency of educational programs, especially those using digital technologies, as well as identify subgroups most likely to benefit from tailored interventions.

## 5. Conclusions

Therapeutic education represents a cornerstone of infection prevention and safety management in rheumatologic care. When implemented within an interdisciplinary framework, it enhances patient awareness, promotes vaccination adherence, and strengthens self-management skills. However, further high-quality evidence is needed to quantify its clinical impact and to guide the integration of innovative, patient-centered educational models into standard care pathways.

## Figures and Tables

**Figure 1 nursrep-15-00431-f001:**
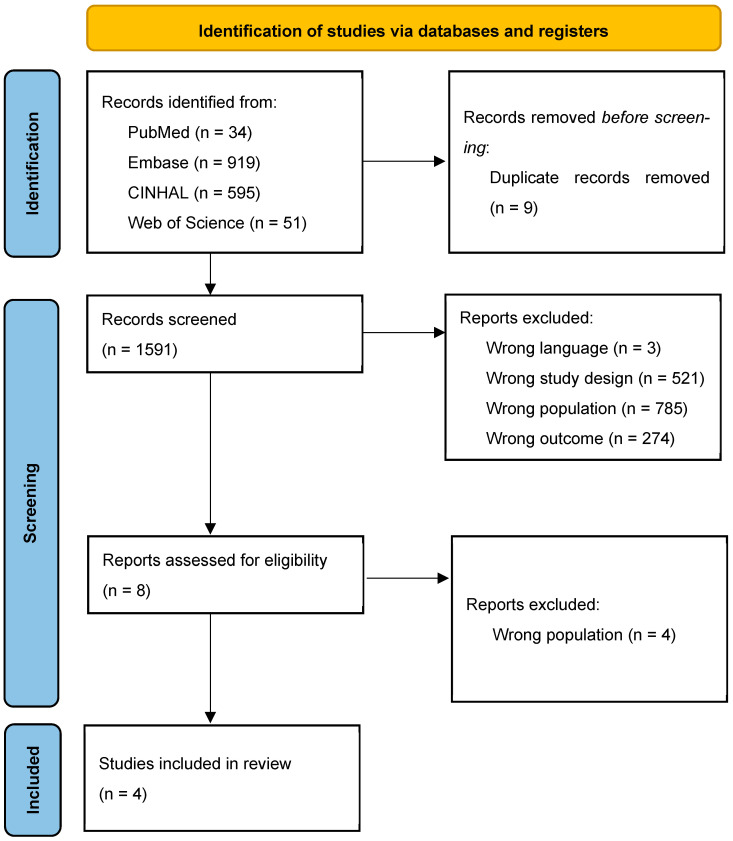
PRISMA flow diagram. Note. The term “wrong” (e.g., “wrong language”, “wrong study design”, “wrong population”, “wrong outcome”) indicates reasons for exclusion based on the pre-specified eligibility criteria described in the Methods section.

**Table 1 nursrep-15-00431-t001:** Methodological quality assessment of included studies using the CASP (Critical Appraisal Skills Program) tools.

Study ID		Q1	Q2	Q3	Q4	Q5	Q6	Q7	Q8	Q9	Q10	Q11	Q12	Tot. of “YES”	Overall Appraisal
**Cohort**															
1	Artime et al. (2020) [18]	Yes	Yes	No	Yes	No	Yes	Yes	Yes	Yes	Yes	Yes	Yes	10	No major concerns
**Cross-sectional**															
2	Kiltz et al. (2021) [19]	Yes	Yes	No	Yes	Yes	Yes	Yes	Yes	Yes	No	Yes	Not available	9	Minor methodological concerns
**Trials**															
3	Beauvais et al. (2022) [20]	Yes	Yes	Yes	No	Yes	Yes	Yes	Yes	Yes	Yes	Yes	Not available	10	No major concerns
4	Kipps et al. (2020) [21]	No	No	Yes	No	No	Yes	Yes	Yes	Not clear	Yes	No	Not available	5	Several methodological limitations

Note. Q1: Was the study’s focus clearly stated? Q2: Was the recruited cohort appropriate to address the research question? Q3: Was the exposure measured accurately to minimize bias? Q4: Was the outcome measured accurately to minimize bias? Q5: Were all important confounding factors identified? Q6: Were confounders adequately considered or controlled in the analysis? Q7: Was the follow-up period sufficiently long? Q8: Was the follow-up rate adequate? Q9: Were the results reported with sufficient precision? Q10: Are the results credible and supported by the data? Q11: Are the findings applicable to the local or clinical context? Q12: Do the benefits of the research outweigh potential risks or limitations?

**Table 2 nursrep-15-00431-t002:** Characteristics of the Studies Included in This Scoping Review.

Author, Year and Country of Publication	Aim of the Study	Study Design	Study Population and Sample Size	Age	Female (%)	Interdisciplinary Team (Healthcare Providers)
Artime E et al., 2020 [18], Five European countries (Spain, United Kingdom, Germany, France, Sweden)	To evaluate the effectiveness of Patient Alert Cards (PACs) for the drug abatacept (ORENCIA) in patients with rheumatoid arthritis by comparing process indicators and clinical and safety outcomes.	Multicenter study consisting of a cross-sectional survey and a retrospective review of medical records.	190 patients.	26–35 years, 6 pts (3.16%); 36–45 years, 21 pts (11.05%); 46–55 years, 29 pts (15.26%); 56–65 years, 55 pts (28.95%); >65 years, 79 pts (41.58%).	146 pts (76.84%)	Rheumatologists and nurses participated in the study.
Beauvais C et al., 2022 [20], France	To evaluate the effect of nurse-led patient education on safety competencies in patients with inflammatory arthritis (IA) treated with bDMARDs.	Multicenter randomized controlled trial	127 patients with rheumatoid arthritis and spondyloarthritis—Group Standard 64, Control Group 63	GC 45.4 ± 13.0; GS 48.6 ± 12.6	GC = 69.8%; GS = 62.5%	Rheumatologists and nurses
Kiltz U et al., 2021 [19], Germany	To evaluate the prevalence of infection, hospitalization due to infections, vaccination status, and infection screening before starting bDMARDs in a cohort of patients with chronic inflammatory rheumatic diseases (CIRD) who received an informative intervention about vaccination strategies.	Single-center cross-sectional study	Population: patients with CIRD; total 975 patients (173 immigrants, 424 with rheumatoid arthritis, 132 with psoriatic arthritis, 145 with axial spondyloarthritis, 41 with SLE); number on bDMARDs = 499	Mean age (±SD): 55.3 ± 15.5 (range 18–90 years)	Female %: RA 67.7%, axSpA 34.5%, PsA 58.4%, SLE 90.6%, Other Diseases 72.6%	Patients most commonly received vaccination information from general practitioners (79%), with rheumatologists providing it less frequently (34%). Other specialists (5%) and public healthcare staff (4%) played minor roles. Overall, 141 patients reported receiving information from multiple healthcare providers.
Kipps S. et al., 2020 [21], United Kingdom	Application of a guide for the rheumatology team to better stratify patients by infection risk, followed by a letter with effective protection and prevention measures against COVID-19 infection.	Cohort of rheumatologic patients receiving DMARDs, biologics, JAK inhibitors, or corticosteroids was stratified by infection risk. Patients with a risk score ≥ 3 (range 0–6) were notified by letter about their risk status and advised to adopt preventive measures against SARS-CoV-2.	Cohort of 887 patients, of whom 248 had a risk score ≥ 3 and received the informational letter on COVID-19 infection risk mitigation.	In the group receiving the letter (risk score ≥ 3), age ≥ 70 years	N.A.	Rheumatology teams (specific healthcare professionals not detailed).

Legend. axSpA: axial spondyloarthritis; bDMARDs: biological disease-modifying anti-rheumatic drugs; CIRD: chronic inflammatory rheumatic diseases; COVID-19: coronavirus disease 2019; DMARDs: disease-modifying anti-rheumatic drugs; GC: group control; GS: group standard; IA: inflammatory arthritis; JAK inhibitors: Janus kinase inhibitors; N.A.: not available/applicable; ORENCIA: commercial name of abatacept; PACs: patient alert cards; PsA: psoriatic arthritis; RA: rheumatoid arthritis; SARS-CoV-2: severe acute respiratory syndrome coronavirus 2; SD: standard deviation; SLE: systemic lupus erythematosus.

**Table 3 nursrep-15-00431-t003:** Key Results and Outcomes of the Studies Included in This Scoping Review.

Author, Year and Country of Publication	Key Findings	Outcome Measures Included	Patient Education Intervention (If Applicable)
Artime E et al., 2020 [18], five European countries (Spain, UK, Germany, France, Sweden).	A statistically significant association was observed between the overall composite score and tuberculosis screening: overall scores ≥ 67%, 34–67%, and ≤33% were associated with TB screening rates of 60.0%, 81.0%, and 54.2%, respectively (*p* = 0.004). No significant correlation was found for viral hepatitis screening (scores ≥ 67%, 34–67%, ≤33% associated with screening rates of 57.5%, 70.7%, and 56.6%, *p* = 0.206). Hospitalization rates for infections increased as patient survey scores decreased: 2.5%, 5.2%, and 8.4%, respectively (*p* = 0.44). No significant correlation was found between questionnaire scores and ER visits or mean time to seek medical care.	Percentage of patients screened for tuberculosis (TB) and viral hepatitis (VH) before treatment; correlation between screening and global patient questionnaire score; percentage of patients with serious infections causing hospitalization or ER visits; mean time between symptom onset and seeking care.	Patient Alert Cards (PACs) provided and explained to patients to raise awareness about infection risks and side effects related to abatacept treatment.
Beauvais C et al., 2022 [20], France	Nurse-led patient education positively impacted safety behaviors in bDMARDs-treated patients; mean biosecure score at 6 months was 81.2 ± 13.1 in the intervention group vs. 75.5 ± 13 in control (difference 5.6, *p* = 0.016), showing improved safety behaviors. Secondary outcomes showed no significant differences except for coping (*p* < 0.03).	AHI (Arthritis Helplessness Index), ASAS, ASDAS, BASDAI, BMQ, DAS28.	Face-to-face patient education sessions at baseline and 3 months focused on safety behaviors and self-injections per French Rheumatology Society guidelines; sessions preceded by individual nurse assessment; supported by booklet; session duration ~65 min at baseline and 44 min at 3 months; both groups continued standard clinical care including medication info from rheumatologists.
Kiltz U et al., 2021 [19], Germany	Patients with chronic inflammatory rheumatic diseases (CIRDs) remain insufficiently protected against vaccine-preventable infections: 7.6% of those vaccinated against measles lacked protective antibody titers. All patients receiving bDMARDs were screened for latent tuberculosis infection (LTBI) and hepatitis B, revealing LTBI in 16 individuals; among HBV-positive patients, 33.4% received prophylaxis and 64.3% showed protective immunity. Only 55.4% presented vaccination records. Although 64.5% (n = 629) had received education on vaccination strategies, adherence remained poor (*p* = 0.200). Despite the availability of professional vaccination counseling, overall vaccination coverage was low-to-moderate. Infection risk (RABBIT Risk Score) did not correlate with vaccination score, but increased age and longer disease duration were associated with higher risk, while low vaccination status correlated with higher disease activity and immigrant background.	Physical function (HAQ, BASFI), Infection risk (RABBIT Risk Score), infection screening, immunization score.	Vaccination information available only via paper-based vaccination cards; no structured educational intervention detailed.
Kipps S. et al., 2020 [21], United Kingdom	Only one patient tested positive for COVID-19 in the high-risk group that received an informational letter, showing infection rates similar to the general population. No evidence that the letter reduced COVID-19 incidence due to lack of control group. COVID-19 incidence was similar between letter group (0.403%) and UK population (0.397%). Trend toward lower incidence (0.113%) observed in the entire cohort.	COVID-19 incidence rates, infection risk assessed via BSR risk stratification guidance.	Informational letter sent to patients with risk score ≥ 3 advising on protective and preventive measures against COVID-19 infection.

Legend. AHI: Arthritis Helplessness Index; ASAS: Assessment of SpondyloArthritis International Society; ASDAS: Ankylosing Spondylitis Disease Activity Score; BASDAI: Bath Ankylosing Spondylitis Disease Activity Index; BASFI: Bath Ankylosing Spondylitis Functional Index; BMQ: Beliefs about Medicines Questionnaire; bDMARDs: biological disease-modifying anti-rheumatic drugs; BSR: British Society for Rheumatology; CIRD: chronic inflammatory rheumatic diseases; COVID-19: coronavirus disease 2019; DAS28: Disease Activity Score on 28 joints; ER: emergency room; HBV: hepatitis B virus; HAQ: Health Assessment Questionnaire; LTBI: latent tuberculosis infection; PACs: patient alert cards; RABBIT: Rheumatoide Arthritis: Beobachtung der Biologika-Therapie (Risk Score); TB: tuberculosis; VH: viral hepatitis.

## Data Availability

Data are available upon request to the corresponding author.

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
