# Peer review of "Therapeutic Education for Safer Rheumatologic Care: A Scoping Review to Map Evidence on Infection Prevention"

_nursrep, 2025, doi:10.3390/nursrep15120431_

Round 1

Reviewer 1 Report

Comments and Suggestions for Authors

I would like to commend the authors for their robust methodological approach, which adheres to the PRISMA consensus guidelines.

The selection process is rigorous, and it is clear that a significant amount of work was invested in reviewing the literature and defining consistent inclusion and exclusion criteria. The final selection of four studies is well justified and clearly presented.

There are, however, a few points that should be addressed to strengthen the manuscript:

  1. Materials and Methods:
    Please clarify whether a “reverse search” (reviewing the references of the included articles to identify additional eligible studies) was performed. If it was conducted, specify how it contributed to the final selection. If it was not, please state this explicitly and provide a brief justification.

  2. Discussion Section:
    The discussion would benefit from a more explicit synthesis of the nursing interventions identified across the included studies. I suggest adding a structured, point-by-point summary outlining each intervention type and describing in which included studies it appeared. This will enhance the clarity and clinical applicability of your conclusions, especially for nursing professionals who may use this review to inform practice.

Overall, this is a valuable and well executed contribution to the field. I look forward to the revised version.

Author Response

Reviewer 1

I would like to commend the authors for their robust methodological approach, which adheres to the PRISMA consensus guidelines.

The selection process is rigorous, and it is clear that a significant amount of work was invested in reviewing the literature and defining consistent inclusion and exclusion criteria. The final selection of four studies is well justified and clearly presented.

There are, however, a few points that should be addressed to strengthen the manuscript:

  1. Materials and Methods: Please clarify whether a “reverse search” (reviewing the references of the included articles to identify additional eligible studies) was performed. If it was conducted, specify how it contributed to the final selection. If it was not, please state this explicitly and provide a brief justification.
  2. Discussion Section: The discussion would benefit from a more explicit synthesis of the nursing interventions identified across the included studies. I suggest adding a structured, point-by-point summary outlining each intervention type and describing in which included studies it appeared. This will enhance the clarity and clinical applicability of your conclusions, especially for nursing professionals who may use this review to inform practice.

Overall, this is a valuable and well executed contribution to the field. I look forward to the revised version.

Response R1 

  1. Thank you for your comment. We have clarified this aspect in the Methods section, specifying that a reverse reference search of the included studies was performed but did not yield additional eligible records.
  2. We thank the reviewer for this helpful suggestion. We added a concise, structured summary of the three categories of educational interventions (informational, communicative, and nurse-led programs), including where each one appeared in the included studies. This improves clarity and the clinical relevance of the findings for nursing professionals.

Reviewer 2 Report

Comments and Suggestions for Authors

I have read the scoping review titled Therapeutic Education for Safer Rheumatologic Care: Mapping Evidence on Infection Prevention with mixed impressions. On one hand, the topic is highly relevant: the review addresses the impact of educational interventions on infection prevention among rheumatic patients receiving DMARDs, a population with an elevated risk of infections and steadily increasing treatment uptake. On the other hand, the findings presented in the review are so heterogeneous that no clear conclusions can be drawn. Consequently, it is unclear to what extent this review meaningfully contributes to the current evidence base on educational strategies for infection prevention in patients receiving DMARDs. I therefore leave it to the editor’s discretion to determine the suitability of this manuscript for publication.

More specific comments are as follows:

  1. The authors restricted inclusion to studies published in English and Italian. It is unclear why Italian was selected in addition to English, aside from the authors’ familiarity with the language. Including only English and Italian while excluding other major languages such as French, Spanish, Arabic, Chinese, Russian, etc. may introduce selection bias..
  2. Please move the current Table 3 to the beginning of the results section and present it as Table 1. In this table, provide a clear summary of the methodological quality of the included studies, in addition to the numerical scores assigned, and indicate whether each study was judged to be of high, moderate, or low methodological quality.
  3. In the current Table 1, the study by Kiltz U. et al. lists a study aim that does not specify the educational strategy being evaluated. This needs clarification.
  4.  Please provide the full list of keywords used during the literature search, including the Boolean operators.
  5. The section on review limitations requires substantial expansion to adequately reflect methodological constraints and the implications of the heterogeneity observed.

Author Response

Reviewer 2

I have read the scoping review titled “Therapeutic Education for Safer Rheumatologic Care: Mapping Evidence on Infection Prevention” with mixed impressions. On one hand, the topic is highly relevant: the review addresses the impact of educational interventions on infection prevention among rheumatic patients receiving DMARDs, a population with an elevated risk of infections and steadily increasing treatment uptake. On the other hand, the findings presented in the review are so heterogeneous that no clear conclusions can be drawn. Consequently, it is unclear to what extent this review meaningfully contributes to the current evidence base on educational strategies for infection prevention in patients receiving DMARDs. I therefore leave it to the editor’s discretion to determine the suitability of this manuscript for publication.

More specific comments are as follows:

  1. The authors restricted inclusion to studies published in English and Italian. It is unclear why Italian was selected in addition to English, aside from the authors’ familiarity with the language. Including only English and Italian while excluding other major languages such as French, Spanish, Arabic, Chinese, Russian, etc. may introduce selection bias.
  2. Please move the current Table 3 to the beginning of the results section and present it as Table 1. In this table, provide a clear summary of the methodological quality of the included studies, in addition to the numerical scores assigned, and indicate whether each study was judged to be of high, moderate, or low methodological quality.
  3. In the current Table 1, the study by Kiltz U. et al. lists a study aim that does not specify the educational strategy being evaluated. This needs clarification.
  4. Please provide the full list of keywords used during the literature search, including the Boolean operators.
  5. The section on review limitations requires substantial expansion to adequately reflect methodological constraints and the implications of the heterogeneity observed.

Response R2 

  1. Thank you for raising this important point. We acknowledge the possibility of language bias arising from restricting inclusion to English and Italian. Our decision was based on practical considerations related to reviewer expertise: these are the languages in which the review team has full proficiency, allowing for accurate screening, data extraction, and critical appraisal. To increase transparency, we have now explicitly acknowledged this as a methodological limitation in the revised manuscript, noting that restricting the search to two languages may have led to the exclusion of potentially relevant evidence published in other languages. We also clarify that such pragmatic language restrictions are common in scoping reviews, where resource constraints and feasibility are recognized considerations under PRISMA-ScR guidance.
  2. Thank you for your suggestion. We agree on the importance of transparently reporting methodological quality. However, the CASP tool does not support numerical scoring nor the categorization of studies into high, moderate, or low quality, as it is designed for qualitative appraisal rather than quantitative grading. For this reason, we summarized the methodological appraisal narratively and reported item-level judgments for each study in the revised table.
  3. Thank you. We have corrected table 1 accordingly.
  4. We have added the full search string in Supplementary Materials as cited in the text.
  5. We agree with the reviewer. We expanded the limitations section by highlighting the lack of standardized outcome frameworks, short follow-up durations, and heterogeneity in study designs, which collectively limit comparability and long-term inference. Furthermore, as this is a scoping review, our objective was to map the breadth of available evidence rather than synthesize effect sizes. The heterogeneity observed is therefore an expected characteristic of the design and has been discussed explicitly in the expanded limitations section.

Reviewer 3 Report

Comments and Suggestions for Authors

The introduction part need to re-do because some statments are not having referebces....e.g line 65-67, line 68-69, line 74-79 
so it is better to recheck the entire introduction part. 

PRISMA flow diagram, the source is missing and also some clerifications needed e.g in it wrong to write the Wrong language (n = 3)
Wrong study design (n = 521)
Wrong population (n = 785)
Wrong outcome (n = 274)
i mean why they considered as the wrong.. ?? need to justify in fig also.
what about the PROSPERO registration need to write in methods.. 
i recomment in Table 1 can please add main findings/summary at the end of each study. 
Table 3. Quality assessment of inlclded studies need to write by using which tool the name is needed here 
Discussion is not at all justified, Need to justify your studies and findings with references and comparison as well 

Author Response

Reviewer 3

  1. The introduction part need to re-do because some statments are not having referebces....e.g line 65-67, line 68-69, line 74-79. So it is better to recheck the entire introduction part. 
  2. PRISMA flow diagram, the source is missing and also some clerifications needed e.g in it wrong to write the. Wrong language (n = 3). Wrong study design (n = 521). Wrong population (n = 785). Wrong outcome (n = 274). i mean why they considered as the wrong.. ?? need to justify in fig also.
  3. what about the PROSPERO registration need to write in methods.
  4. i recomment in Table 1 can please add main findings/summary at the end of each study. 
  5. Table 3. Quality assessment of inlclded studies need to write by using which tool the name is needed here 
  6. Discussion is not at all justified, Need to justify your studies and findings with references and comparison as well 

Response R3

  1. Thank you for your comment. We have included references for each statement.
  2. Thank you for your comment. In the PRISMA flow diagram, the term “wrong” (e.g., “wrong language”, “wrong population”) was used only as a concise label to indicate that full-text articles did not meet one or more pre-specified eligibility criteria. We have now clarified this by adding an explanatory note to the figure note specifying that “wrong” refers to reasons for exclusion based on the predefined inclusion/exclusion criteria reported in the Methods section.
  3. Thank you for this observation. This study was conducted as a scoping review, and according to PRISMA-ScR guidance, protocol registration is not mandatory. In addition, PROSPERO does not currently accept scoping review protocols for registration. For these reasons, the review was not registered.
  4. Thank you for this suggestion. We agree that summarizing the main findings of each included study is valuable for readers. To avoid excessive complexity in Table 1, we created a dedicated table (now Table 3) reporting a structured summary of the main findings for all included studies. We have now clarified this in the manuscript.
  5. Thank you for your comment. We have now specified in the title/legend of Table 3 that methodological quality was assessed using the CASP (Critical Appraisal Skills Program) tools (now Table 1).
  6. Thank you for this valuable suggestion. We strengthened the Discussion by integrating additional references and more explicitly comparing our findings with existing literature in chronic disease education and infection prevention. This allows clearer interpretation of how the identified interventions align with broader evidence supporting interactive, patient-centred educational strategies.

Round 2

Reviewer 2 Report

Comments and Suggestions for Authors

The authors have addressed all comments except for one regarding the rationale for including studies published in English and Italian. The current explanation provided by the authors remains rather vague. Therefore, I leave it to the editor’s discretion to determine how best to address this point.

Author Response

Thank you for your comments which have helped improve our manuscript.

Reviewer 3 Report

Comments and Suggestions for Authors

I had gone through the changes; only one change has still not been incorporated. In the discussion part, line 345-46 (few studies have quantified the effect of education....) no reference added I mean don't know which studies the author is refereeing actually. Please check that. 

Author Response

Thank you very much for your comments, which have strengthened our manuscript. We have clarified that the term “few” refers to the studies included in our scoping review, as well as to the content of that entire paragraph.